# Recent Advancement of Data-Driven Models in Wireless Sensor Networks: A Survey

**Gul Sahar** [1,2,*] **, Kamalrulnizam Abu Bakar** [1] **, Sabit Rahim** [2] **, Naveed Ali Khan Kaim Khani** [3] **and Tehmina Bibi** [4]

1. Faculty of Engineering, School of Computing, Universiti Teknologi Malaysia, Johor Bahru 81310, Malaysia; knizam@utm.my
2. Department of Computer Sciences, Karakoram International University, Gilgit Baltistan 15100, Pakistan; sabit.rahim@kiu.edu.pk
3. Department of Electrical Engineering, Federal Urdu University Arts Science and Technology, Islamabad 44000, Pakistan; naveedali64@yahoo.fr
4. Institute of Geology, Faculty of Science, University of Azad Jammu and Kashmir, Muzaffarabad 13100, Pakistan; tehmina.bibi@ajku.edu.pk
* Correspondence: gulsahar@kiu.edu.pk

**Abstract:** Wireless sensor networks (WSNs) are considered producers of large amounts of rich data. Four types of data-driven models that correspond with various applications are identified as WSNs: query-driven, event-driven, time-driven, and hybrid-driven. The aim of the classification of data-driven models is to get real-time applications of specific data. Many challenges occur during data collection. Therefore, the main objective of these data-driven models is to save the WSN's energy for processing and functioning during the data collection of any application. In this survey article, the recent advancement of data-driven models and application types for WSNs is presented in detail. Each type of WSN is elaborated with the help of its routing protocols, related applications, and issues. Furthermore, each data model is described in detail according to current studies. The open issues of each data model are highlighted with their challenges in order to encourage and give directions for further recommendation.

**Keywords:** wireless sensor network; application types; data-driven models; event-driven; query-driven; time-driven; hybrid-driven; data collection; energy consumption

## 1. Introduction

Wireless sensor networks (WSNs) have recently evolved as a very vigorous research area in the field of advanced network communication. Moreover, it also plays an important role in terms of rapid technological progress, emerging practical development, and application activities [1]. The rapid growth of WSNs between 2012 and 2022 has been reported as increasing from 0.45 billion to 2 billion [2]. The major advantages of WSNs are self-organized and configured within the specified time interval, which is fixed by developments in manufacturing. Furthermore, the high impact of WSNs is enhanced in the capacity, integrity, and reliability of the network. Mostly, the WSNs are composed of many tiny and scattered sensor nodes, which have less battery capacity, thereby raising the issue of power and energy consumption. The fundamental purpose of WSNs is the collection of data by the collaboration of intermediate sensor nodes via the wireless connection. Due to any changes that occur in the environment such as temperature, humanity, velocity, speed of the wind, and light, data need to be updated according to the application requirements. Some areas including building monitoring, smart agriculture, and healthcare are easy to deploy, while the environmental [3,4], glacier [5], habitat and traffic monitoring [6], hazardous chemical detection, earthquake [7], volcano eruptions and disaster management [8], and weather forecasting and flood detection [9] are not considered as an easy

deployment due to their hostile environment. Moreover, in these environments, the sensor nodes are distributed randomly by dropping through the uniform airdrop deployment method (UAD).

The contributions of this survey article are mentioned as follows:

- The WSNs types in different groups are classified according to their protocols, applications, and current issues.
- The data-driven models for WSNs categorize into four groups such as query-driven, event-driven, time-driven, and hybrid-driven.
- These data models are described according to the related issue and their proposed solution.
- Last, this survey also highlights each data model's limitations and challenges for helping new researchers to work on new enhancements and modification in the field of data-driven WSNs.

The rest of this paper is organized as follows: Section 2 consists of a summary of the earlier existing studies, surveys, and reviews from the last six years on WSNs data-driven models. Section 3 describes in detail the WSNs' architecture overview. Section 4 presents a detailed taxonomy of WSN types in different groups according to their protocols, applications, and current issues. In Section 5, the detailed description of WSNs data-driven models is presented related to data management for energy efficiency. Section 6 highlights each data-driven model's limitations and challenges for helping new researchers, and the conclusion is presented in Section 7.

## 2. WSNs Architecture Overview

A typical WSN's architecture consists of a sensing field that contains various kinds of sensor nodes such as data-sensing nodes, the base station, and the sink, all of which are related to each other. In the sensing field, the base station collects the data from all of its sensor nodes and transfers these data to the sink node. The sink node collects the data from all the sensor nodes efficiently due to its high processing, functionality, and memory [10]. Each sensor node is composed of different basic components, namely a sensing unit, microcontroller, wireless transceiver, and battery by determining the physical objects. After the acquisition of the sensing unit, the data are digitalized by using a microcontroller, processing power, and coordination capability [11]. Now, the data are transferred into the other connected sensor nodes through the transceiver. The transceivers are used for the transmission and reception of the data. WSNs are divided into four types based on the kind of sensor node: static, mobile, hybrid, and mobile robot [12]. Normally, all deployment of static wireless sensor networks (S-WSN) sensors is static, but mobile wireless sensor networks (M-WSN) sensors, such as i-Mouse, are fitted with locomotive platforms and may move after the first installation. Another side, a hybrid wireless sensor network (H-WSN) integrates both static and mobile sensors. Instead of implementing mobile sensors, robots are utilized to handle static sensors; these networks are known as WSRNs. Robots within wireless sensor and robot networks (WSRNs), also known as carrier-based, contain the static sensors as payload and install these sensors in suitable areas. Therefore, in this scenario, several of the predetermined deployment models are adopted while the robot's mobility is optimized using a route optimization algorithm [13].

Finally, battery power is the main component that is used in all steps of data communication through the data process, and users can easily access data and information according to need through the internet [14]. The typical network architecture of sensor nodes in WSNs is shown in Figure 1.

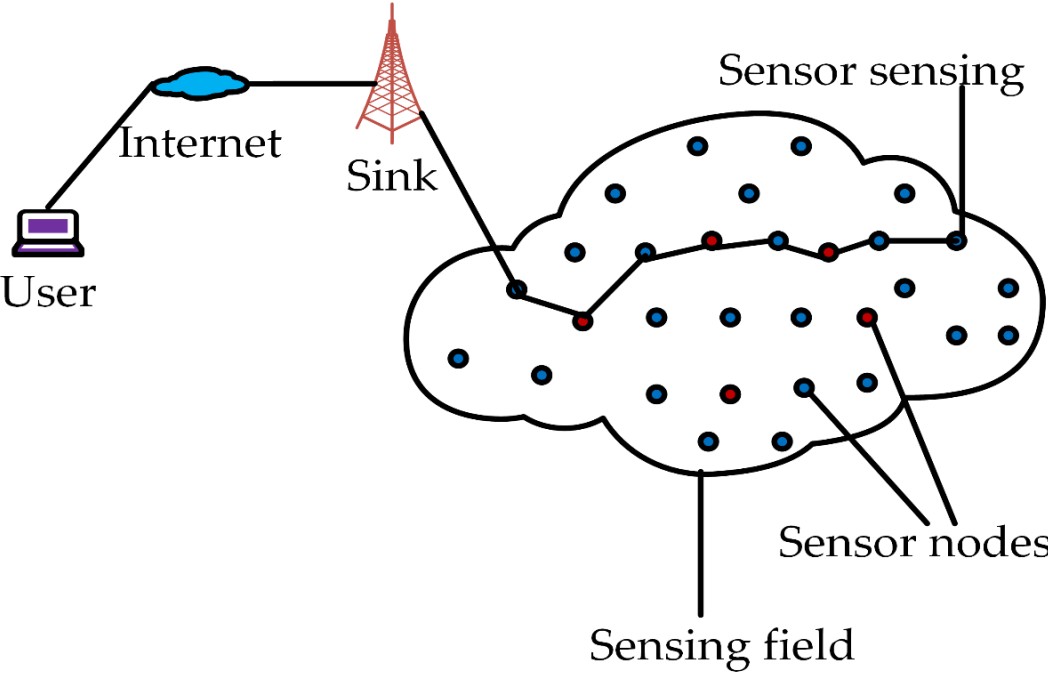

**Figure 1.** The typical network architecture of sensor nodes in a WSN.

The data transmission is transferred from one location to another location through the routing protocol in WSNs. A well-designed WSN protocol helps the network live longer. For this reason, the proposed protocol is made with considerably more emphasis than the other aspects of design. The WSN protocol architecture is designed to be energy efficient and meet application needs. There are two types of data collection protocols now in use: hierarchical protocols and non-hierarchical protocols [15]. The attraction of a non-hierarchical collection of protocol modules over a completely vertically stacked design, which is common in several other groups of communication systems, is that events transmitted through one layer (module) do not need to be analyzed with an intermediary layer. Nodes are drawn to the sink node because it avoids constraints and is maintained within the transmission range. Non-hierarchical WSN deployments are usually best for an event-driven data model, but hierarchical architecture implementations are better for long-term deployments. The hierarchical routing architecture gives higher coverage, while the geographic routing implementation delivers an extended network life. It may seem prudent to adopt a hybrid method, where the most computationally intensive tasks are done by a power mote [16].

Chain, tree, grid, and cluster-based hierarchical routing are the four primary kinds of hierarchical routing. Chain-based routing is simple to install and manage; the chain's structure does not vary frequently, and nodes always transmit data to the closest node. As a result, the amount of energy required to build a chain is minimal. The drawback in chain-based routing is that there may be various nodes inside a chain; however, if the sensor node is far from the sink node, the data must travel a long distance to reach the sink node. The delivery period is considerable, which may create significant delays and make it unsuitable for time-sensitive applications [17]. The nodes are split into numerous branches, leaf nodes, and parent nodes in tree-based routing. Data are sent to the leaf node toward its parent node, next to another parent node, and so on, until they reach the sink node. The disadvantage of tree constructions is that when a tree's parent node fails, all data transfer within its branches would be lost. Furthermore, if indeed the branch has a large number of sensor nodes, data transfers are delayed, and energy demand may rise [18]. The whole network space is split into several grids, each with its own cluster head. All sensor nodes

in a grid will transmit data to its cluster head, which will transmit it to the cluster head of another grid until it arrives at the base station [19].

Unfortunately, when a grid has a large number of sensor nodes, perhaps there is a lot of data traffic, and the cluster head node's energy is depleted quickly. The tree, chain, and grid-based routing have significant issues in that long-distance communication is not possible due to the lack of scalability for various sensors. Network clustering is attempted in various routing techniques to address these disadvantages. The cluster-based method divides networks into several clusters. Each cluster is made up of several sensor nodes, one of which is chosen as the cluster head. Due to the decreased number of data packets to be delivered, the bandwidth is kept to a bare minimum. In the cluster-based technique, the data-aggregating method decreases the quantity of immediately transmitted data to the base station as well as the amount of energy that is required due to the shorter transmitted power [20].

## 3. Related Work

In this section, a brief review of the various existing studies and survey literature about WSNs is presented. WSNs design is a huge challenge to sustain because the geographical structure includes small-scale, large-scale, and hostile areas. In this geographical structure, various applications are used that have their own requirements and related issues. There are only a few studies that focus on WSNs data-driven models. Hence, this survey presents the application types, architectures, data-driven models, and challenges/limitations with future recommendations for data-driven models in WSNs, as shown in Table 1.

**Table 1.** Comparison of existing related works of data-driven models in WSNs.

| References | Applications | WSNs Types | Architecture | Data-Driven Models | Challenges/Limitations |
|---|---|---|---|---|---|
| [4] | Yes | No | No | No | No |
| [10] | Yes | No | No | No | No |
| [21] | No | Yes | Yes | No | Energy efficiency, topology design, cost, antenna design, condition and type of soil, variable requirements, environment size, and underground |
| [22] | No | Yes | No | No | Security, computational and memory, hardware design, cost, and power consumption |
| [23] | Yes | Yes | No | No | Resource constraints, quality-of-service, security, data redundancy, packet errors, variable-link capacity, and storage |
| [24] | Yes | Yes | No | No | Energy constraint, transmission media, computational capability, limited bandwidth, fault tolerance, scalability, and cost of deployment |
| [25] | Yes | | Yes | Yes | Delay, network size, energy-efficiency, and scalability |
| [26] | Yes | No | Yes | No | Node weight and dimensions, robustness, communication range, throughput, reliability and security, network tolerance |
| [27] | Yes | | Yes | | Resource constrains, communication cost, streaming data, heterogeneity and mobility of nodes, communication failures, large-scale deployment, identifying outlier sources |
| [28] | Yes | Yes | No | No | Limited bandwidth, delay variance and propagation, transmission range, complex acoustic environment |
| [29] | Yes | Yes | No | Yes | Data collection and storage, data processing |
| [30] | Yes | Yes | Yes | No | Battery power issues, communication issue, severe environment conditions |

**Table 1.** *Cont.*

| References | Applications | WSNs Types | Architecture | Data-Driven Models | Challenges/Limitations |
|---|---|---|---|---|---|
| [31] | Yes | No | Yes | No | No |
| [32] | Yes | Yes | No | No | Fault tolerance, scalability, transmission media, power constraint, management at a distance sensor, and security issues |
| This Survey Article | Yes | Yes | Yes | Yes | Yes, present in Session 6 |

Table 1 shows the difference between this survey article and existing studies in the area of WSNs for data management. Each WSNs type along with related applications is explained in detail [23,24,32]. The WSN types and architecture are presented in [21,31], data aggregation and non-data aggregation are outlined in [22], and WSNs' applications are explained with the increase in network lifetime in [4,10]. Energy-efficient routing protocols are designed for WSNs applications in [32]. Hence, this survey presents the WSNs' applications, types, architecture, and data-driven models in detail. After the configuration of WSN types, applications, and architecture, data need to be captured and delivered by sensor nodes. Data are based on different data-driven models according to the WSN scenarios. Data-driven models are necessary for handling the data in WSNs to enhance network lifetime, congestion control, traffic control, and overload, while data redundancy reduction, packet loss, data delay, and data accuracy affect each data-driven model. Thus, the researcher needs to do more work for the enhancement of data-driven models.

A top–down survey [4] presents a battery system between application requirements and the increase in the battery life while designing WSNs. The main categories of applications are healthcare, the environment, agriculture, industry, transportation, public safety, and military system. The requirements of each application are stated in the survey as well. However, battery replacement is very pricey and impractical, especially in hostile areas. The existing standard of low power includes Zigbee, wirelessHART, ISA100.11a, IEEE 802.15.6, Bluetooth low energy (BLE), and 6LoWPAN. These standards do not respond to all applications. The survey also reviews major energy-saving mechanisms, discusses their advantages, and suggests further recommendations.

Elimination of data redundancy, reduction in network traffic, and enhancement of WSN lifetime are some of the significant techniques of data aggregation. Various existing types of data aggregation techniques and protocols are surveyed [21]. The data integration technique with clustering in WSNs is presented in detail for the types of underground, underwater, and ground sensor networks. Terrestrial WSNs are based on structure and structureless methods that are influenced by data aggregation architecture with computational intelligence. The underground WSNs entail different challenges for the networks such as energy efficiency, topology, dense application, and power communication reduction. Additionally, we briefly introduce the two underground WSNs architecture such as static and mobile UWSN. Wireless body sensor networks (WBSNs) of physical components of sensor nodes, architecture, and data aggregation protocols are also explained.

The survey [22] provides an overview of WSNs and their types. A basic block diagram of a node, the physical parameters of WSNs, and design challenges such as energy consumption, cost, hardware design, computational, security, memory cost, and specific environment are described in detail. The types of WSNs presented include the mobile WSNs, underwater WSNs, underground WSNs, wireless multimedia sensor networks, and terrestrial WSNs. Similarly, a state-of-the-art review has presented the design and key challenges of WSNs. The current and attractive main requirements and constraints for WSNs and a general review of WSNs applications and their types are elaborated in detail in [23].

WSNs applications diversity is rapidly increasing due to the wide range of interactive communication in the surrounding nodes. The study focused [10] on the analysis and evaluation of different classification and characterization parameters for WSN applications because researchers and designers need more help to identify and satisfy the specific WSNs applications. The characterizations of various parameters are classified into six categories: communication and traffic, node, network, the operational environment, service component, and service.

The structure of the open system international (OSI) model for the protocol stack consists of three layers. On the other hand, the international standard organization (ISO) proposed seven layers. Note that the WSNs' architecture consists of only five layers, and each layer has different characteristics, as presented in [24]. The study reviews WSNs' characteristics with advantages and challenges, applications, topologies, and types.

For the designing of WSNs, different applications are used and consist mostly of factors such as objectives of applications, challenges, cost, hardware, and environmental condition. The remote sensing applications are presented with an expanded survey [33] to identify new and existing applications. The survey recommends both system requirements as well as the protocol stack of WSNs in parallel, while classifications with different challenges and issues are faced in various environments.

A comprehensive survey of hierarchical routing protocols that are constructed for mobile wireless sensor networks is presented in [25]. The purpose of the study is to propose a routing protocol that supports sensor node mobility in mixed WSNs, which include static and mobile sensor nodes. Furthermore, the researchers emphasize the benefits and drawbacks of each routing approach as well as performance concerns. The study focuses on discussing some of the most current traditional and efficient hierarchical routing that has already been constructed. The survey additionally includes a comprehensive categorization of the examined methods based on various parameters. The routing approach, mobile element, control manner, mobility pattern, clustering attributes, network architecture, protocol operation, communication paradigm, path establishment, protocol aims, energy mode, and applications are mentioned as performance metrics. Furthermore, we investigate the protocols based on latency, network size, energy economy, and scalability, as well as the benefits and limitations for every protocol.

The purpose of the paper is to investigate several problems of interest linked to WSNs by looking at specific instances of actual WSN applications, both commonly used and innovative ones [26]. The use of WSNs in various domains, such as heathland, military, urban, flora, wildlife, and industrial, is investigated in this article through the examination of relevant representative cases that are both innovative and well known. Additionally, the author's investigation revealed that the utilization of WSNs not only gives numerous benefits in selected domains especially compared to the earlier comparative methods and techniques, it also presents innovative applications. Furthermore, both the challenges and solutions produced for a wide variety of applications are highlighted and reviewed.

The author of [27] presents a review of detecting outliers in WSNs. The research also includes details on WSN applications and descriptions of outliers from earlier research. Furthermore, several categories of outlier origins in WSNs were explicitly explained. The focus of the research has been to give a complete analysis on WSN outlier detection. The research provides a variety of approaches and investigates outlier detection strategies in each application area as well as its benefits and drawbacks. Furthermore, the limitations of outlier methods in WSNs are also discussed. Finally, the study evaluated the specified approaches for outlier diagnosis in WSNs from the perspective of its properties, applicability, and limitations.

Similarly, the aim of the research in [28] is to investigate the area of underwater wireless sensor networks and give a thorough understanding of UWSNs' objectives, architectures, current advancements, taxonomy, and limitations. Furthermore, the study presents the most recent data for several factors that can fulfill the criteria for speedy UWSN growth. Additionally, the research presents a review of literature using well-known

and reliable article databases. These categories help pinpoint future problems for enhancement and provide more possibilities for long-term contribution to the area of underwater sensor networks.

WSNs have a wide range of potential applications. The application of such networks in the context of big data highlights its capacity to transcend fundamental limitations in order to satisfy specific objectives. In another study, [29] presents two main points regarding big data collection in LS-WSNs. To begin, the capability of WSNs and big data is explored, with a focus on data collecting. Next, current data techniques are examined and debated in the research. The article provides an overview of big data gathering in LS-WSNs and also the limitations that must be handled. Furthermore, the aim of the study is to give research and industry with important insights into its potential field of study as well as to encourage the development of innovative big data collection in LS-WSNs schemes and structures.

In the study, Xu, G et al. [30] focuses on the specific application of WSNs in marine environment monitoring because of its easy deployment, real-time observing, automatic process, and low cost. The study provides a current state-of-the-art review of WSN applications for marine environment monitoring. It starts by going through the basics of WSN-based marine environment monitoring, such as possible applications, a basic WSN architectural, a generic sensor node design, sensory parameters and devices, and wireless communication technologies. Then, the study goes over the relevant study in terms of numerous studies, methods, methodologies, and strategies for monitoring the maritime environment using WSNs. According to the results of the study, there are still a few significant problems and possibilities in the development and implementation of WSNs for maritime environment monitoring, such as sensor security and stability.

## 4. Types of Wireless Sensor Network

In this section, the wireless sensor networks (WSNs) are classified into six types, including terrestrial, underground, underwater, multimedia, mobile, and wireless body area networks according to the environment. Therefore, each WSN type has its specific data-driven model mode that is based on data capture involving computation, analysis, and transmission method. The relationship between network application types and data-driven models, as in all land-dwelling applications, are known as the terrestrial type of WSN [34]. A fire in the forest is an example of an event-driven model. When fire occurs in a forest, sensor nodes sense and detect the specific data (event-driven model), collect the data, and send data to the sink node. Second, weather forecasting is a form of periodic or continuous (time-driven model), where sensor nodes capture the data continuously and send it to the sink node according to the time slot. Third, the query-driven model is mostly used in a healthcare environment where the operator creates the query for the specific requirement of the data in the healthcare process. Finally, hybrid data depend upon the environmental situation and condition of the scenarios. More than one data-driven model can be used in a specific application. For example, in an industrial environment, continuous data that are used for monitoring and observation are known as time-driven data, while during breakdown or some emergencies, these data are known as event-driven to resolve the issues or event by sensor nodes.

Each group has its own various routing protocols, application, and related issues and constraints. Figure 2 shows the WSNs' various types, applications, architectures, data models constraints, and challenges.

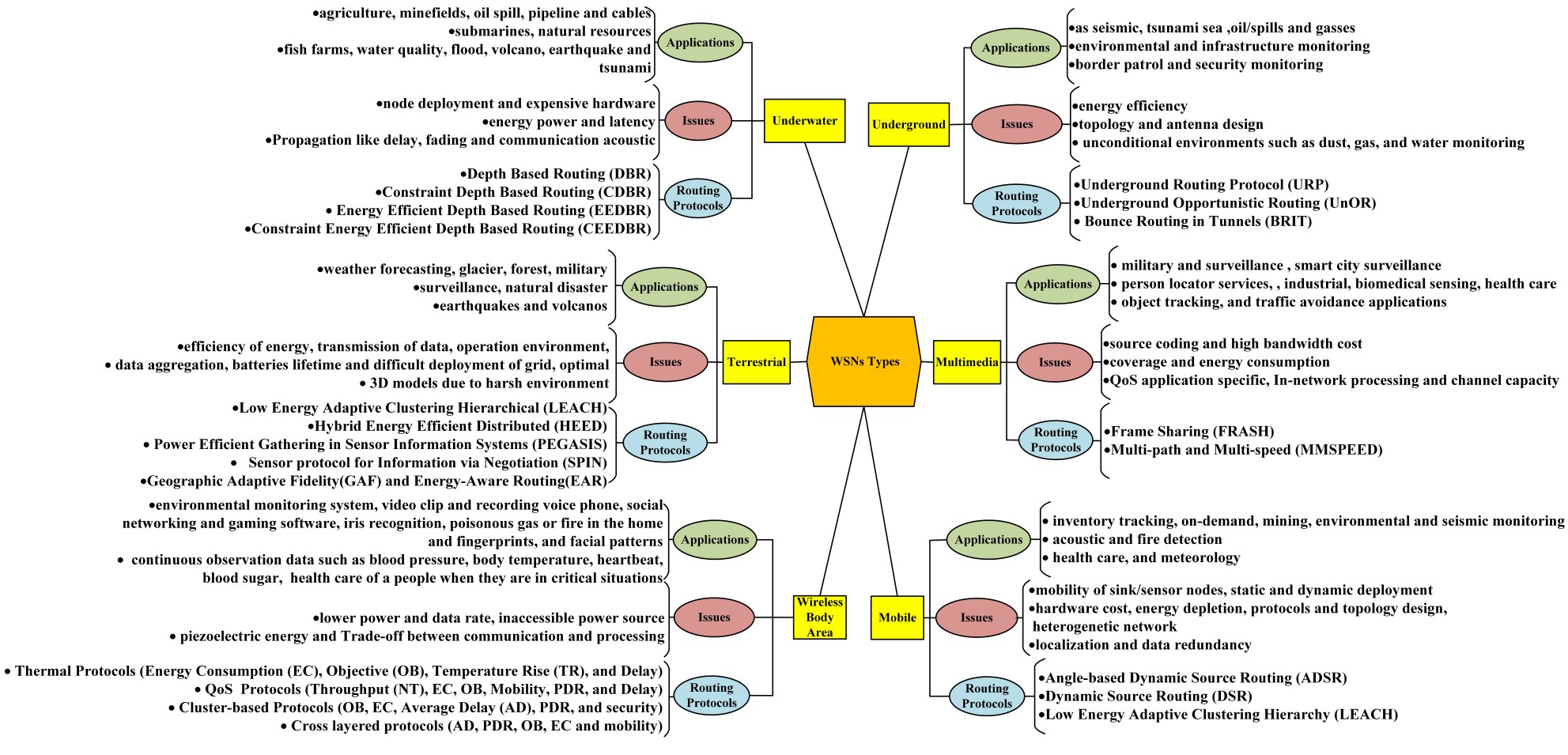

**Figure 2.** WSNs various types, applications, architectures, data models constraints, and challenges.

### 4.1. Terrestrial WSNs

Terrestrial WSNs are land-dwelling spaces containing hundreds of thousands of sensor nodes deployed in interesting places. These nodes are very low cost or inexpensive. Two types of terrestrial WSNs are deployed such as unplanned and pre-planned (sensor nodes are arranged in 2D or 3D placement) [35]. These nodes are driven by battery power. Due to the harsh or hostile environmental condition of nodes deployment, nodes are not rechargeable and not replaceable [33]. Energy preservation is one of the hardest challenges [10] in terrestrial WSNs. In addition, energy consumption within in-network communication is one of the most critical challenges due to data redundancy. The data aggregation techniques are used for energy conservation at each sensor node during data collection. Therefore, energy can be preserved through multi-hop optimal routing, short transmission range, data aggregation, eliminating data redundancy, and delays [36]. Low-energy adaptive clustering hierarchical protocol (LEACH), hybrid energy-efficient distributed routing (HEED), geographic adaptive fidelity (GAF), power efficient gathering in sensor information systems (PEGASIS), sensor protocol for information via negotiation (SPIN), and energy-aware routing (EAR) are routing protocols that are mostly used. The main terrestrial application is weather forecasting, glacier, forest, military surveillance, natural disaster, earthquakes, and volcanos. The major challenges and issues are the efficiency of energy, transmission of data, operation environment, data aggregation, battery lifetime, difficult deployment of the grid, and optimal and 3D models due to the harsh environment for terrestrial WSNs. The major challenge of the continuous data stream in WSNs is to get valuable data by deployed sensor nodes due to with limited battery power. The skyline query is the one of popular query-driven models; [37] study snapshot skyline query on a data stream by sensor nodes in WSNs. The query-driven model is used for environmental monitoring and event surveillance application, which are considered in the terrestrial WSNs application type.

There are various complex monitoring applications of terrestrial WSNs; for example, fire detection requires smoking density and temperature observation data; security requires vibration and infrared radioactivity data, and many sensitive areas where people are not allowed and people are tracked by privacy emergence monitoring need an event-driven data model [38].

Time-driven data models are used for time-series data collection in periodic wireless sensor networks. The time-driven data model is divided into periods, and the further periods are divided into slots with respect to fixed time slots on every sensor node. Time-driven data applications generate large volumes of data by the sensor nodes due to this high energy consumption and reduced network lifetime [39].

### 4.2. Underground WSNs

Underground WSNs have also been considered terrestrial WSNs in the last decade. In this network, two types of wireless devices are considered, such as on-ground transceiver devices, while these devices avoid communication under the soil. In the soil sensor device, these are generally connected to gain access to data [40]. However, underground WSNs require a high cost to design, implement, and maintain the special sensor for the network communication. Mostly, application sensors are deployed under the ground for agriculture and minefields to monitor conditions in the soil, environment, infrastructure, border patrol, and security monitoring [41]. Recently, underground applications consist of agriculture for large-scale environmental monitoring to improve farm choice and to detect accidental infection such as soil monitoring, underground environments, and levels of water information under the surface [42]. On the other hand, WSNs are used to monitor underground environments such as mines and miners' security to transfer information in case of a disaster. However, the most difficult function is to certify safe working situations in coal mines underground where unconditional environments such as dust, gas, and water monitoring exist [43]. Underground WSNs have design routing protocols for their specific

applications such as underground routing protocol (URP), underground opportunistic routing (UnOR), and bounce routing in tunnels (BRIT) [44]. Some existing studies examine underwater WSNs for pipeline and mine monitoring [45,46]. A time-driven data model such as temperature readings from the pipe's environment and the pipe surface has the ability to provide crucial data for the leakage's identification and localization. When water leaks from a pipe in to the underground, the local temperature profile in the ground might vary when compared to values measured at other nodes far away from the leak [45]. Similarly, event-driven data models are used for safety, of which coal mine productivity is a top focus, and the model performs an essential part in it. Coal mining is mostly done underground, when geological structures are more difficult. As coal mining progresses to deeper levels, the volume of gas emitted rises, increasing the danger of coal and gas outbursts. Other natural causes, such as rock breaks, coal dust bursts, and water leaks also create tragedies in deep mines. WSNs have the potential to observe and analyze changing, hostile, and unfamiliar situations. As a result, a mesh WSN with multi-parameter monitoring in underground coal mines is created, functioning as a replacement to the cable monitoring system (CMS), depending on advanced technologies [46].

*4.3. Underwater WSNs*

Underwater WSNs are extended terrestrial WSNs that have attracted considerable attention because of their value in obtaining resources that are difficult to transport. Underwater WSNs are deployed in the sea. However, its hostile environment due to the time taken for exploration, expense, and people is difficult to measure [47]. UWSNs and terrestrial WSNs can be compared based on their different environment variable quantities. However, a UWSN has high signal attenuation because underwater communication is acoustic, with frequent synchronization, propagation delays, limited memory, and less bandwidth. These sensor nodes cost more than terrestrial sensors [48,49]. Some UWSN applications include seismic and tsunami sea life exploration, environmental condition of underwater quality, oil/spills, gas monitoring, agriculture, minefields, pipeline and cables, submarines, natural resources fish farms, flood, volcano, earthquake, coalmine tunnels, and tsunamis [50]. The most critical problem of UWSNs is the energy consumption of sensor nodes, because it is more difficult to recharge or replace battery power. Thus, energy saving to prolong the network's lifetime becomes a crucial issue in many UWSN applications [41]. Efficient routing protocols have been developed for UWSN. Such protocols are constraint depth-based routing (CDBR), which is an extension of (DBR) [51], and constraint energy-efficient depth-based routing (CEEDBR) extended from (EEDBR) [52]. There are four types of UWSN applications: scientific, industrial, military, and security. Monitoring, controlling, and ocean sampling in specific sectors have great barrier reef operations and are among the many uses for UWSNS in the scientific sector. Furthermore, time-driven data are collected by robotic fish to monitor temperature and pressure as well as detect oxygen levels in water [53]. UWSNs' industrial applications have a major influence on commercial activity facilitation. Underwater monitoring applications of oil and gas pipeline are possible with UWSNs. In the study [54], a prototype is designed for monitoring underwater oil and gas pipelines. The technology is created to give an event data-driven model of the health of pipelines that operate across huge territories. Similarly, Ross et al. [55] also built an underwater oil and gas pipeline monitoring system, which is required for the monitoring of an actuator mechanism. The various military and security applications utilize event-driven data models such as performing port and harbor control and monitoring, sea mine identification, border security from unauthorized battleships or submarines, and decentralized situational surveillance, military, and defense applications, which employ a mix of underwater sensors to look for possible threats early [56,57].

*4.4. Multimedia WSNs*

Multimedia is a new and emergent type of WSNs. The real-time environmental scalar data, images, video stream, audio data, and tracking movement are only retrieved with

multimedia WSNs. Multimedia networks consist of sensor nodes furnished or fixed with microphones, cameras, and other multimedia objects. These types of networks have additional abilities such as the processing time, storage of associates, and fusion of multimedia data from different sources [58]. The aim of the multimedia network is a wide range of possible applications such as military and surveillance [59], smart city surveillance [59], person locator services and environmental monitoring [60], industrial, biomedical sensing, and healthcare [61], object tracking, and traffic avoidance applications [62]. The big challenges in multimedia sensors are source coding, high bandwidth cost, coverage, and energy consumption, QoS application specific, in-network processing, and channel capacity [63]. Two main types of routing protocols are used such as frame sharing (FRASH) and multi-path multi-speed (MMSPEED). One of the event-driven data applications in wireless sensor networks with the most potential is movement tracking. However, existing scalar-based WSNs are incapable of extending various image data, such as object kind and shape. To solve the complexity, a multimedia WSN is used to handle the object's different data. When adopting multimedia, service quality issues always emerge, necessitating the use of network technologies to provide consistent service quality. In the study [62], the authors show that the quality of service (QoS) for multimedia applications varies depending on applications. As a result, delivering QoS while optimizing a network becomes a major issue. Event-driven and query-driven data-driven models are the two most common data models of basic service modes. The service mode based on events only has one sort of service, which is an event-driven service [64]. The majority of event-driven data model applications in multimedia sensor networks are latency and error intolerant. Monitoring the technological ensemble from nodes is essential for an application. Query-driven models have two sorts of services available, including the data query service and stream query service. The majority of services for data inquiry are aimed at the error-prone but query-specific delay-tolerant applications [65,66].

### 4.5. Mobile WSNs

Mobile WSNs is the most recent type where mobile sensor nodes always change their positions and restructure the networks. Mobile WSNs are also called hybrid networks due to their combination of fixed and mobile nodes. When sensor nodes are deployed, information on the field of interest is gathered [49,67]. To increase the mobile WSNs' mobility, the number of applications is scattered by people, animals, manned vehicles, auto vehicles, and unmanned vehicles. Mobile WSNs' applications are widely classified into inventory tracking, on-demand, mining, environmental and seismic monitoring, acoustic and fire detection, healthcare, and meteorology [68]. The main issues and limitations of the mobile WSNs are the mobility of sink/sensor nodes, static and dynamic deployment, hardware cost, energy depletion, protocols and topology design, heterogenetic network, localization, and data redundancy. The routing methods of mobile WSNs are divided into hybrid, distributed, and centralized, while two types of routing approaches are used, which are classical-based and optimized-based. In mobile WSNs, the mostly used existing protocols for communication are angle-based dynamic source routing (ADSR), dynamic source routing (DSR), and low-energy adaptive clustering hierarchy (LEACH) [69]. In a mobile WSN (MWSN), the easier option is to use stationary sensor nodes and move the sink nodes. A query-driven data model used for crops on a farm may contain sensors that detect humidity or temperature, and anytime a farmer passes by, its smartphone functions as a sink node, while allowing the data to be downloaded. When sensor nodes are connected to animals in tracking applications, event data are detected by the static sinks while sensor nodes are mobile. When the animals are within its range, a static sink is utilized to gather tracking data stored in the sensor nodes [70]. On the other hand, target coverage is critical since it is prevalent in applications. In these scenarios, in data gathered, wherever only the data on certain points need to be gathered where the event occurs. Since sensors are generally distributed in a messy manner throughout a large area, moving a subset of mobile sensors is frequently necessary to guarantee adequate coverage of the

sensing field [71]. Unmanned aerial vehicles (UAVs) are widely used for data collecting and picture capture in current history. UAVs are used with wireless sensor networks (WSNs) to develop data-collecting systems with a wide range of capabilities. These methods are designed as UAV-WSNs, but they are ideally adapted for remote monitoring and emergency response, particularly in the region of landslides, wildfires, floods, and other disasters applications. Event-driven data model observation is the foundation for making the appropriate judgement in these circumstances. Additionally, event-driven data detection still saves the lives and cost to a certain measure [72].

### 4.6. Wireless Body Sensor Networks (WBSNs)

Considered as one of the most enabling technologies, WBSNs are mostly used in medical and non-medical applications. The non-medical applications include environmental monitoring system, video clip and recording voice by a mobile phone, social networking and gaming software applications, iris recognition, poisonous gas or fire in the home, and fingerprints and facial patterns. Medical applications include continuous observation data such as blood pressure, body temperature, heartbeat, and blood sugar. It is a subfield of WSNs due to its continuous monitoring; the main intention is the healthcare of people when they are in critical situations. The intelligent biosensors are used as a sensor device in wireless BSNs, which are fixed outside or inside the human body to sense the interesting information from the human body; then, sensory data are transferred to the base station for further processing. Wireless BSNs are widely used in many applications such as healthcare, medical and non-medical, and telemedicine. The main challenges faced in wireless BSNs are low power, low data transfer rate, inaccessible power source, piezoelectric energy, and the trade-off between communication and processing. The routing protocol in Wireless BSNs is classified in various groups such as thermal routing protocols (energy consumption, objective, packet delivery ratio, temperature rise, and delay), QoS routing protocols (throughput, energy consumption, objective, mobility, packet delivery ratio, and delay), cluster-based protocols (metrics, energy consumption, objective, average delay, packet delivery ratio, and security), and cross-layered routing protocols (average delay, packet delivery ratio, energy consumption, objective, metrics, and mobility) [73]. WBSNs must satisfy the quality of service (QoS) demands of users/applications and the relevant network. In order to describe QoS criteria, users must first determine the type of applications that need to be used as well as their data delivery models from sensors to the base station, such as continuous/time driven, query driven, event driven, or hybrid driven. Each of these data delivery model types has its own set of requirements for quality of service. Every data delivery model has its own QoS criteria for QoS metrics except for the continuous/time data delivery model, while the other data delivery models must meet all four essential requirements: real-time delivery, reliable delivery, energy efficiency, and flexibility to network channel circumstances [74]. The continuous data-driven paradigm usually withstands minor delays and packet losses. While there is a high rate of packet loss on a communication connection, QoS structures or methods must take extra provisions, for events such as a heart attack delivering crucial data. It ensures that the important data reach the destination in a timely and accurate manner [75].

## 5. Analysis of Data-Driven Models for WSNs

In WSNs, the data transmission between sensor nodes and the sink node is delivered by data-driven models according to the nature of the data. The data-driven models are used for specific interest objects which are classified into four basic data-driven models in WSNs, as shown in Figure 3. These models, such as query-driven, event-driven, time-driven, continuous-driven, and hybrid-driven [76,77], are described in detail in the following subsection and presented in Table 2.

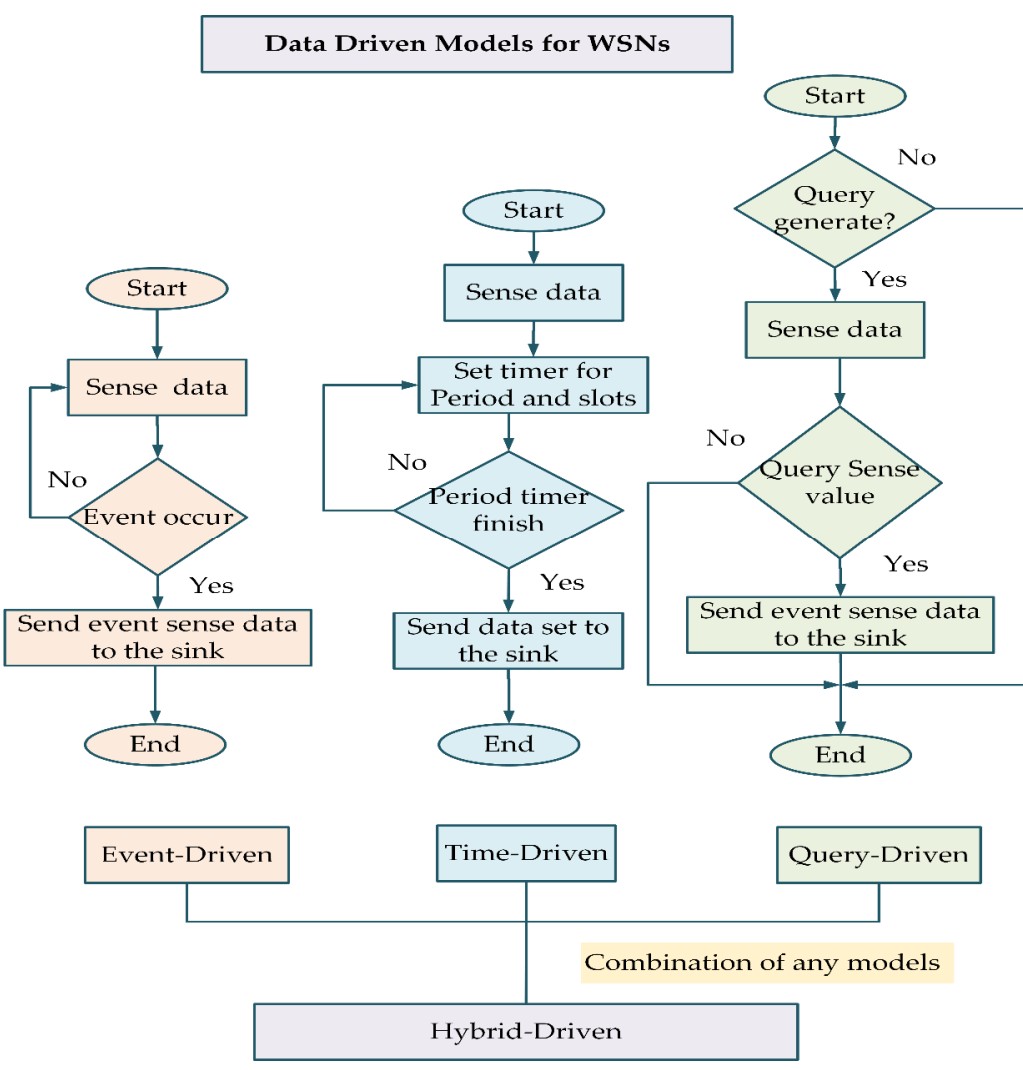

**Figure 3.** Process of data-driven models for WSNs.

**Table 2.** Summaries of existing studies for data-driven models in WSNs.

| References | Applications | Data Models | Problem Identify | Network Lifetime | Energy Wastages | Data Delivery | Data Aggregation | Methods | Limitations |
|---|---|---|---|---|---|---|---|---|---|
| [78] | Health-monitoring applications | Query driven | Data collection WSNs | Multiple trees | Parent node | Delay requirements are not | - | Sleep cycle scheduling | Need real-time applications such as battery life of nodes, critical response time, mobility nodes, and a lifetime of WSNs |
| [79] | Post-emergency management system | Query driven | Quality of service (QoS) | Energy consumes | Wheel nodes | Data delivery delay | Wheel node performs data aggregation | Wheel maintenance mechanism | Does not support multiple sink nodes and latency of the application |
| [80] | Emergency operations, battlefield environment | Query driven | Network control overheads quality of service | Reduced network congestion | Cell-header/forwarder nodes | Data delivery performance | Cell header | Progressively shifting cell-header | Does not handle random data propagation and volatile sink mobility patterns |
| [81] | Fire temperature | Event driven | End-to-end delay | Event-driven clustering | Active state for gather data | Latency increases quickly | Cluster formation based lead to redundancy | Sleep state | Do not focus on multi-source data and multi-path fusion |

**Table 2.** *Cont.*

| References | Applications | Data Models | Problem Identify | Network Lifetime | Energy Wastages | Data Delivery | Data Aggregation | Methods | Limitations |
|---|---|---|---|---|---|---|---|---|---|
| [82] | Wildfire, earthquake, high chemical density and flood | Event driven | True or fault event | Residual energy | Forwarder node | Link reliability delivery ratio (DR) | Distributed event detection | Minimum hop-count | Focus only on a specific environment |
| [83] | Monitoring an event of interest | Event driven | Overhead of mobile sink location | Shortest routing path | Transmitting and receiving data | Minimizes data delivery delay | Reducing the overhead of advertising | Virtual wheel-based data dissemination | Data transmission link can be lost due to the presence of an obstacle in the sensing environment |
| [84] | Grounded robots or vehicles environment | Event driven | Predefined trajectory environment | Mobile sink predefined trajectory | Fixed trajectory environment | Data delivery time | Rendezvous points (RPs) | Number of hops sensor-node data transformation | Do not handle a lifetime of network and energy consumption |
| [85] | Environmental monitoring forest, wildfire | Event driven | Energy hole problem | Reducing the number of hops | Sensors nearby the sink run out of energy | - | Abnormal event | RP planning | Do not consider data redundancy to save energy consumption |
| [86] | Production efficiency of factories, monitoring and controlling individuals' city data | Time driven | Spatial and temporal correlation | Faulty nodes | Whole sensor nodes | - | At the node level and the cluster head level | Matching-weight-enabled aggregation (EMWA), marginal weight-enabled aggregation (MWA) and Euclidean | Data sense collection is long due to this large information loss |
| [87] | IoT data health in a community, | Time driven | Outliers and redundant data | A load of network and energy consumption network lifetime | Dense sensor nodes, network congestion, and traffic | - | Cluster Head | Cosine similarity function, Mahalanobis distance | Need data classification for accuracy |
| [88] | Temperature, humidity, light, and voltage | Time/periodic driven | Reliability and information | Data redundancy or similarities | Each node level and CH level | - | Redundancy data | Aggregation, compression, and predictions | Do not tackle data collision and scheduling |
| [89] | Pit mining | Time driven | Data aggregation of heterogeneous data | Heterogeneous data collection | Data transmission, sensor nodes, and a central node | - | Central node | Average, sum, minimum, and produce a packet | Deplete sensor batteries quite fast due to the high amount of data redundancy |
| [90] | Home, industrial, logistics, aviation, health, manufacturing, and military | Time driven and event driven | Data prioritization of heterogeneous | Data priority class for energy saving | All the sensors | Packet delivery ratio | - | Class of service traffic priority-based medium access control (CSTP-MAC) | Still, data handling of data prioritization for the network layer of the WSN protocol stack |
| [91] | Temperature, light intensity, magnetism, seismic activity, sound, water level | Query driven and event driven | Data query dissemination scheme and data gathering | Energy threshold changes | All the sensors | Data delivery rate and latency | - | Hybrid data dissemination protocol (HDDP) | Do not consider QoS (Quality of Service) parameters and the mobility of some nodes |
| [92] | Coffee Arabica and Coffee Robusta | Time driven and event driven | To detect time-driven or event-driven data | Energy expended on cluster-head | Data communication from sensor node to a base station | - | Each node level | Cluster-based data aggregation (CBDA) | Event-driven model accuracy true event information |

**Table 2.** *Cont.*

| References | Applications | Data Models | Problem Identify | Network Lifetime | Energy Wastages | Data Delivery | Data Aggregation | Methods | Limitations |
|---|---|---|---|---|---|---|---|---|---|
| [93] | Security specifications | Hybrid is driven (event and time monitoring) | Non-time-critical and time-critical monitoring | - | - | Accuracy for event or time monitoring | Constraints on sequences of action | Authentication mechanisms | It is not actually designed for secure software |
| [94] | Healthcare monitoring | Query driven | High reliability | Low | Packet lost collusion, over emitting | Prime requirement | No need | Sleep and wake up | High memory cost |
| [95] | Advertising mobile sink's location by flooding | Query driven | Injection location and data collection location | Select minimum length routing path | Queries and data routing | Data delivery delay | Rendezvous nodes | Virtual ring infrastructure | Data loss if one sensor node fails |

The data processing of each data-driven model is different. Suppose the query-driven model starts the process with query generation for getting data from sensors as per the user requirement. Once the user generates queries, then, it sends the generated query to the WSNs' sensor nodes. If the query matches with the data of the sensor nodes, then it is accepted; otherwise, it is rejected and deleted. For the event-driven model, sensor nodes are mostly in sensing mode. Sensors usually sense data according to an event, and at the moment the event occurs, data are sent to the sink node; otherwise, data are sensed continuously. In a time-driven model, sensors also sensed the data continuously. The sink node sets a timer to all sensor nodes for sensing the data. Sensor nodes divide that specific time into periods and further divide periods into slots. The sensor nodes send the data to the sink within a specific period of time; otherwise, sensor nodes collect the data continuously. Lastly, the hybrid model is made from merging two models. Suppose if the hybrid model is made by merging event-driven and time-driven models. Normally, the hybrid model processes with respect to the time-driven model to sense and send the data periodically. Whenever an event occurs, the hybrid model switches from a time-driven model to an event-driven model for further processing in the WSNs.

*5.1. Query-Driven Model*

In WSNs, a query-driven model is used when the user needs data regarding their requirement. According to the user's need, the user sends the request to the sensor node in the interested region. Interested regions have various types of nature such as environmental monitoring, agriculture, healthcare, military, forest monitoring, etc. Hence, there is a huge problem regarding energy in WSNs due to the harsh environment. The energy problem in this model does not affect the whole network; it only affects the nodes of a specific region, which is consumed by data transmission.

For reducing energy consumption, Snigdh et al. [78] propose a data protocol based on multi-tree and multiple-tree algorithms. In tree-based architecture, the parent node has more load as compared to the child node, as the child node forwards data as well. The parent node receives data from the child node as it might have redundant data, which can cause high transmission. Mostly, data load occurs due to congestion, packet loss, collusion, chances of data being delayed, network control overheads with quality of service (QoS), and loss of reliability. The query-driven model is usually used for fast communication; most of the applications include health monitoring applications, post-emergency management systems (rack any object), and fire temperature.

Jain et al. [79] propose a QWRP: a query-driven virtual wheel-based routing protocol for location-based data collection. The data collection is by a mobile sink node that is placed in an interesting location-based region. To improve data delivery performance, QWRP is used. In the network, special nodes are known as wheel nodes, which have the information of mobile sink nodes. When the mobile sink wants to get the data from the interested region, wheel nodes send the query to the interest region of the mobile sink node

with the help of an angle-based forwarding algorithm. The main purpose of the wheel node algorithm is to decrease the data delivery time and enhance the data delivery ratio. However, more wheel node energy is wasted compared to simple network nodes.

In the same way, Khan et al. [80] propose a query-driven virtual grid-based data dissemination QDVGDD protocol for wireless sensor networks consuming a single mobile sink to decrease the network overhead and control the data delivery ratio. Initially, WSNs constructed a virtual infrastructure that controls and spreads both query and response packets. A mobile sink surrounds the sensor field in a clockwise direction and then spreads query packets in the sensor field to show its interest.

*5.2. Event-Driven Model*

Event-driven model applications are used for emergency and disaster recovery-based applications such as health emergencies, forest fires, earthquakes, monitoring of air quality, animal movement, rain, lava eruption, military applications [96], and volcanic eruption [97–99]. The main feature of the event-driven model is that the collection of data is not formed on a continuous, regular basis while an event is occurring. An event-driven model requires reliability and assurance of delivery on time due to the emergency in the specific region. In the same environmental conditions in which the event occurs, region nodes collect and transmit redundant or raw data, which causes a high transmission rate. The raw form or redundant data causes serious event issues to occur as the region's sensors face congestion, data redundancy, and network overload, as high traffic generates high energy consumption and high transmission cost.

In WSNs, the data fusion with Dempster–Shafer evidence theory and event-driven algorithm (EDDS) are proposed to reduce the data size and energy consumption [81]. The EDDS algorithm is used on a group of nodes, and the data sampling rate is based on the threshold when an event occurred. Furthermore, weighted data fusion is used for practical confidence and structure rules when data are transmitted from cluster heads (CHs) toward the sink. EDDS detects anomaly data and finds the data redundancy to identify the occurrence of event.

An event-driven routing between event and fault disambiguation is a very big challenge. Biswas et al. [82] present a true event-driven and fault-tolerant routing (TEDFTR) algorithm for event-driven routing. When an event occurs, any node sends events occurrence information report to the base station through multi hopping with the help of the TEDFTR algorithm. TEDFTR is a distributed event detection algorithm that detects true environmental events such as earthquakes, floods, high chemical density, and wildfire. To assure that the event is true, fault measures between neighbor nodes are carried out through a voting system. To relay is selected based on the hop-count and multi-object weight sum method through the next-hop node for an event alert.

In WSNs, updating the mobile sink's node's location is a very big challenge for high communication overhead and high energy consumption. An event-driven virtual wheel-based data dissemination (EDVWDD) scheme is proposed for sensor fields to include multi wheels to decrease the mobile sink node updates [83]. Whenever an event occurs, general sensor nodes send event-sensed data to the mobile sink. With the help of EDVWDD, a new virtual wheel structure is constructed where the event occurs. The nodes in the virtual wheel structure access the mobile's sink location easily while the remaining nodes do not send data toward the mobile's sink. When under the virtual wheel structure, nodes receive information about the mobile's sink and then send their data to the sink with the help of a geographical routing algorithm for the shortest path selection and data delivery delay minimization.

Vajdi et al. [84] investigate the problems of hostile environment applications or event-driven wireless sensor networks (EWSNs) such as forest and fire detection events. In these applications, data harvesting is a big challenge to handle from sensor nodes in the field. These applications are used in many physical and arbitrary trajectory obstacles where the mobile sink cannot easily move. The predefined trajectory is based on EWSNs in the

environment where there is no possibility to visit each sensor node one by one. Thus, sensor nodes are deployed randomly by aircraft. The study aims to select a set of rendezvous points (RPs) with a predefined trajectory for two main purposes, which are less energy consumption and the control of high data delivery delay in the EWSNs. A direct influence on the lifetime of WSNs is found on energy consumption is directly related to the rate of data transmission.

Similarly, Zhang et al. [85] propose an entropy-driven data aggregation with gradient distribution (EDAGD) deployment strategy for maximizing WSNs lifetime. The EDAGD strategy contains three algorithms. A multi-hop tree-based data aggregation (MTDA) framework is proposed for minimizing path distance, which decreases data transmission distance in the transmission process by limiting the number of hops required. An entropy-driven aggregation tree-based routing algorithm (ETA) is suggested for the Choquet integral and entropy, which uses data aggregation to monitor abnormal events. During the data transmission process in abnormal areas, sensor data are set to function, and the remaining nodes are set to sleep to save energy. For reducing the energy hop problem, a gradient deployment algorithm (GDA) is used.

### 5.3. Time-Driven Model

A time-driven model is a sequence of data points composed of overtime intervals by a track system based on time. In WSNs, time-driven data are produced via a sensor node by a periodic form from environmental conditions. The environmental condition does not affect network sensor nodes, these sensor nodes are continuously sensing and sending data to the sink node. The sink node collects the data from the sensor nodes in a form of a round base on time duration. Now, each sensor node is divided into the period, and each period is divided into fixed time slots that represent specific reading. Time-driven networks sometimes experience the physical observed environment condition that features dynamic change slow down and speed up. As a result, a lot of redundant or raw data products that cause various types of traffic are generated in WSNs such as packet losses, high energy consumption, data representation, data accuracy, data redundancy, network overload, high transmission cost rate, data latency, congestion, data delivery ratio, etc. In recent years, various studies have been proposed to tackle these challenges and issues [86–89].

In WSNs, data redundancy is a challenging issue that arises due to temporal and spatial correlation data collection from sensor nodes. Roohullah Jan et al. [86] propose two new lightweight fold aggregated techniques such as exact matching-based weight-enabled local data aggregation EMWA, and marginal weight-enabled local data aggregation MWA at the node level. For exact matching and to remove marginally aggregated data, EMWA is used to reduce redundancy by similar and average functions. The main aim of EMWA is to reduce the average data transfer rate. To eliminate data redundancy, data with similar values are removed and weighted values are used. To achieve high data accuracy, integrity, and reliability, EMWA uses exact matching data values instead of approximate matching values. MWA aims to calculate the distance between two consecutive readings with a predefined threshold value for identifying marginal differentiation. These extremely small values with updated weighted values are stored in memory and consume energy.

In contrast, outlier detection with an efficient data aggregated scheme is proposed for a fundamental process of saving energy and prolonging network lifetime in WSNs [87]. The novel data aggregated scheme is a radial basis function for data aggregate RBFDA based on cosine similarity and neural network that observes dense areas in a cluster neighborhood together with sensor nodes due to high similar data achievements. The neighborhood member nodes calculate data by the cosine similarity function for the detection of similarities. The Mahalanobis function uses the data outline detected from multivariant data at the node level. The CH node receives a data set from the member nodes and then checks data determination and classification, deletes outlier and reduces data redundancy based on multivariant values for improving data accuracy, and decreases network overload, all of which save energy save prolong the network lifetime.

Ibrahim et al. [88] propose an All-in-One hybrid data gathering and energy-saving mechanism for WSNs. The main idea of their mechanism is a self-reconfigurable sensor based on various parameters to select the mechanism that is suitable for data reduction techniques such as data redundancy amount and remaining power battery. The All-in-One mechanism is divided into three phases. In the first phase, the aim is an on-period data reduction technique to decrease the data transmission amount from sensor nodes by any aggregation, prediction, or compression. In the second phase, based on in-period data adaptation, the rate variation considers the monitoring condition for data transmission by using on–off transmission and adaptive frequency sensing. Third, in-node means the data correlation among the neighboring nodes by using data clustering techniques for in-network correlation.

A large-scale WSN includes many sensor nodes and thus faces issues in terms of data analysis and gathering; it has some limitations regarding data redundancy, sensor energy, and network lifetime. Ramezanifar et al. [89] propose an open-pit mining data aggregation technique for heterogeneous clustering sensor networks. Heterogeneous sensor data aggregation is difficult as compared to homogeneous data. The packet ID is used to identify different packets created by several sensors and various applications. The mining pit technique captures as several similar packets as needed, aggregate them strategically and dynamically, and then transmits them to the base station node. To reduce transmission and consume less energy, packets are sent by single hop near neighboring nodes with the help of the mining pit method.

*5.4. Hybrid-Driven Model*

A hybrid-driven model is a type of data collection and transmission from the sensor nodes that combines aspects of three data models: event-driven, query-driven, and time-driven. When an event occurs, the sensor collects and sends data of the event that is occurring only. After that, once the event stops, then nodes collect the data normally or periodically. On the other hand, when the user sends or requests query-driven according to their need, the sensor nodes send the data on the request of the user through a query. In WSNs, hybrid model sensors have the right to choose the data-driven model according to the occurrence of the event by modification of sensor processes. In existing studies, some researchers combined the methods and schemes to build hybrid models.

Onwuegbuzie et al. [90] propose a class of service traffic priority-based MAC (CSTP-MAC). In the work, data are classified into two classes; the first class is high-priority data (HPD) with priority tag 0 similar to a normal continuous data-driven model, and the second class is low-priority data (LPD) with priority tag 1 similar to an event-driven data model for a wireless body area network (WBAN). The body sensor monitors the body blood, temperature, heartbeat rate, and oxygen level where data collection on various body parts for monitoring prioritizes the patient's most critical conditions. The hybrid data are used in two different ways, such as critical and non-real-time data. Both data have various data communication processes, traffic for low and high data, and bandwidth usages.

The hybrid data dissemination protocol (HDDP) uses two data-driven models proposed by Guerroumi and Khan Pathan [91]. The event-driven model detects the data; when an event occurs, the sensor nodes send data to the sink through an optimal path for energy consumption, and latency increases. The query-driven model is used when a user requests a sensor node through a source in the network.

Similarly, WSNs are used in a fixed and specific field called coffee production. An efficient technique cluster-based data aggregation (CBDA) for a hybrid model is proposed [92]. A hybrid model is composed of two models such as a time-driven and an event-driven model for less energy consumption in data transmission. The time-driven model sensors collect data from the fields and aggregate them into a constant time and next forward them to the base station. In an event-driven model, if any event occurs, data are aggregated by data aggregation technique and sent to a base station. The data aggregated CWSB technique uses cluster-based, either post event detected or after a specific time expiry.

A hybrid monitoring of software design [93] presents security specifications that are divided into two types for observed monitor objects. The first type is event monitor (inline) and the second is the time monitor (offline). Most of the time-static structure and dynamic behavior are used to build a hybrid monitoring model for enhancing the performance and advantages. Time monitoring is used for critical time limitations, while non-critical limits are used for event monitoring in hybrid data models. Monitoring software is used to monitor both the time and event, to execute out on a log result.

WSNs are made up of various individual sensor nodes that may connect to several other nodes for collecting data from the surrounding environment. The hostile environment conditions necessitate the development of WSNs that can be dropped with minimum danger from airplanes, where sensor nodes immediately begin to collect data and send it back to the base station for further analysis. In order to communicate, sensor nodes send the data to the base station, by using intermediate hoping in terms of neighbor collaborations. Thus, energy saving is a major issue in WSNs because the lifetime of sensor nodes depends on battery power.

Existing studies propose several techniques for saving energy in WSNs such as [100] hierarchical clustering, duty cycling and data-driven approach, rendezvous algorithms, [101] mobility-based energy conservation schemes, energy efficient sleep scheduling in [102,103] data reduction, protocol overhead reduction, and topology control. In Table 2, various healthcare applications are used in the query-driven model in terms of data delivery delay, and scheduling algorithms are used to save energy in WSNs [78–80,94,95]. If an unexpected event occurs in the WSNs environment such as fire, temperature, or a road accident, then emergency applications are used in the event-driven model [81–85]. Sensor nodes use the shortest minimal hop routing path for data while analyzing the reliability and true or fault probability-based events that occur.

The time-driven model emphasizes natural phenomena such as light, wind speed, temperature difference, rain, etc. Sensor nodes continuously collect and transmit data, which increase energy cost. High energy cost has a harmful impact on the sensors lifetime due to redundant data, data congestion, high traffic data, and network overload. For less energy consumption, data aggregation, clustering, data redundancy reduction, and routing protocols are used in WSNs [86–89]. Previous studies used a combination of time-driven and event-driven (hybrid) models. Sensor nodes handle two types of data, which are critical and non-critical data, which are classified on a priority basis. If a time-driven model is used, then non-critical data are sent; if an event occurs, the priority goes to the event-driven model. Most event-driven models are based on priority in the hybrid-driven model. Authentication methods, cluster-based data aggregation, hybrid data dissemination protocol, and a class of service traffic priority-based mechanisms are proposed to resolve the issues of data accuracy, data delivery rate, and latency [90–93]. However, many issues and challenges need to be resolved by future researchers related to data-driven models according to the WSNs types, application, and architectures.

Table 2 presents various existing studies and the analysis that the current work has done, and we identified the problems with respect to all four data-driven models and used applications regarding the performance metrics such as network lifetime, energy wastage, data delivery, and data aggregation in WSNs. As query driven data models are used for healthcare applications to handle the issues of data quality and data reliability, the results improve the data delivery ratio and delay. All these processes are carried out by using sleep/active scheduling algorithms. Event-driven data models in previous work target emergency applications such as wildfire, fire temperature, earthquakes, and road accidents. The true or false event direction and end-to-end delay problems are solved by increasing the latency and data delivery ratio. Query responses on data delay can be made by using mechanisms such as sleep/active scheduling and a virtual ringing wheel.

On the other hand, the time-driven data model in earlier work focuses on environmental monitoring applications such as temperature, humidity, light, pet mining, and pressure. The data redundancy, temporal, and spatial correlation issues are solved for

decreasing the data size, reducing the transmission cost, and enhancing energy efficiency in WSNs by using mechanism such as data aggregation and data reduction methods. Similarly, the hybrid data model in previous work focuses on security specification, industrial environment, healthcare, and coffee arabica applications, whereas the two data-driven models are combined to detect the data including the fire, face detection, and temperature. If the time-driven model is used in one of the hybrid models, then it removes the data redundancy and increases the energy efficiency in WSNs. When the event occurs, then it gives priority for event data collection, so that it can quickly estimate the critical event and reduce reporting delay. Mechanisms such as hybrid data dissemination protocols, data aggregation, and classes of priority are used in the hybrid data model. In the next section, various important issues and limitations are highlighted with respect to data-driven models for further research.

## 6. Limitations and Challenges of Data-Driven Models

In this section, limitations and challenges for all data-driven models in WSNs are highlighted for the future research:

*Query model*: In a network, a query is used for the specific information of a place. For a request from the user, it is important to have geographical knowledge of the sensors' location. To complete a request, the wireless communication links, topology, and deployment of sensors should not have inconsistency. To receive a query and send it back, sensors choose the single best route path from all the other paths. The biggest challenge that the query model faces is that there are many obstacles in large area wireless deployment; such obstacles are roads, mines, outdoor hostile areas, and pipelines. When large spaces occur among the sensor nodes, transmission cost increases, as collision occurs due to the small space from node to node.

*Event model*: In this model, data are highly important at times when a specific region of an event occurs. For data accuracy, high-density sensors node deployment is required. However, in most high-density deployment, event-occurring sensors send redundant data, which causes energy consumption. The node coverage must be good for event detection. The biggest challenge is the transmission link between the sensor nodes and the sink. The event information must deliver in the shortest time with a high probability to reach the sink with the exact location.

*Time model*: In the time model, the biggest challenge is energy consumption due to hostile areas as well as a large environment where humans cannot replace the sensor's battery easily. The fault in link communication between transceiver and receiver latency is not correct, and packets drop due to the high transmission cost. Researchers should focus on enhancing the network lifetime, which directly reduces data redundancy and affects data accuracy. Mostly, data delivery is based periodically in real-time applications. Data delivery latency depends upon radio connectivity and is used for delivering data in multi-hop communication. The communication between transceiver and receiver is decreased by using multi-hop, which affects the latency and energy of sensors.

*Hybrid model*: Many limitations arise when a sensor switches its data from one model to another, as there are some difficulties during data processing and communication. A sensor's life depends on the energy and battery lifetime. In hybrid data models, every data model uses different operations for data storage and sending. To identify the sensor's location, the data delivery ratio, accuracy, and high transmission cost are limitations of the hybrid model.

Table 3 shows the limitations and challenges based on data-driven models in WSNs. For future work, the construction and enhancement of a new and innovative routing protocol are demands for the data query model to save the energy of the sensor nodes of the routing path. The elimination of data redundancies is also required for the data route. Moreover, time-sensitive scenarios in any query to further improve data delivery delay problems are required to be resolved. In an event-driven model, various protocols and schemes are required to solve the data delivery delay, end-to-end delay, data accuracy,

latency, and reliability issues. This is because when an event occurs, the recognition of true or false event information is needed for a quick and safe response.

**Table 3.** Analyzes the limitation and challenges of each data-driven model in WSNs.

| Data Model | Limitation and Challenges | | | | | | | | | |
|---|---|---|---|---|---|---|---|---|---|---|
| | Energy Consumption | | | Transmission Cost | Data Delivery Delay | End to End Delay | Accuracy | Reliability | Mobility | Latency |
| | Node | CH | Network Lifetime | | | | | | | |
| Query driven | high | medium | low | medium | medium | medium | medium | medium | high | Medium |
| Event driven | medium | medium | low | medium | high | high | high | high | high | High |
| Time driven | high | high | high | high | medium | medium | medium | low | low | Low |
| Hybrid driven | high | high | high | high | high | high | high | high | high | high |

Furthermore, the time-driven model considers the data redundancy cause of high transmission and network lifetime at the sensor node level; thus, the cluster head level still needs to improve for energy consumption and data accuracy. Similarly, in the hybrid model, time-driven, event-driven, and query delivery-related issues and challenges need to be the focus of future researchers. In addition to these issues, switching between models, prioritization, authentication, handling the critical and non-critical data, storage of data, and data processing and analyzing issues need to be improved and focused on for the further research.

Table 3 is based on Table 2 for analyzing the performance metrics in WSNs that help the researchers with further research, and the future works related to each data-driven model are described in detail.

- In the time-driven data model, the network lifetime is considered as a highly critical issue. The dense deployment sensor nodes are in a hostile monitoring environment for continuous data collection. When any node fails to perform the specific functions, hardware risk occurs due to the changing condition of the surroundings such as overcooling and overheating. Therefore, there is a need to increase the network lifetime so that the model can work in harder conditions such as with glaciers and at high temperature in industrial environments to enhance the network's functionality.
- Additionally, there are various implementations of WSNs for continuous data collection in hostile environments. In this scenario, several sensor nodes are moved far away from the wireless connections in the base location. Hence, these nodes depend upon the entire network of sensor nodes for data transfer to the base station, so that there is a need to work on these out-range sensor nodes for the data collection and location identification.
- The event-driven data model is mostly used when an event occurs in any location of WSNs. A lot of primary data are lost because certain threshold values are set to detect an event so that there is a need to recover the lost data during the initial stage of an event.
- In certain areas, when an event occurs, only active nodes transmit the data. Due to high data transmission, the nodes' energy is imbalanced across the whole network, which also causes high energy consumption.
- Various applications based on irregular or fault events sent information to the sink nodes such as in busy traffic where no accident has occurred but the data are interpreted as there being a road accident; all these problems could be improved in future work.
- The query-driven feature in WSNs: only the sink node generates queries for the whole network due to the high communication cost, many queries could collect a similar data response by various sensor nodes in the network. However, due to double communication (forward and backward query), the network's performance became slow and weak.

- Investigation of the interaction of a cross layer and double communication with routing and query processing still needs to be explored in future work.
- A hybrid data-driven model is considered more challenging and limiting than other data-driven models. Suppose a hybrid model is based on time and event-driven models. Mostly the data redundancy is removed, and the energy efficiency is enhanced by using the time-driven model. If any event occurs during this time, then priority is given to the event-driven model to resolve the issue. Hence, there is a need to save the data loss during the control moving from one model to another model in WSNs.
- There are various challenges to face related to the hybrid data model in WSNs; future research recommendation include data splitting loss, high data changes, data monitoring, data handling, data analysis, updated information, location identification, and data transmission delay.

## 7. Conclusions

The impact of data-driven models is used to enhance the reliability of data or information for WSNs. This survey article provides WSN types with their protocols, issues, and related applications. Data collection models are elaborated in four categories: event-driven, query-driven, time-driven, and hybrid-driven according to their specific descriptions and use in recent studies. Most of the data models are based on data collection, which considers the energy, data latency, data transmission data accuracy, and data delivery ratio for enhancing the energy in WSNs. A comprehensive assessment of the open limitations of each data model with their challenges is highlighted in order to encourage and give directions for future research.

**Author Contributions:** Conceptualization, writing—original draft preparation, review and editing, G.S.; supervision, K.A.B.; investigation and visualization, N.A.K.K.K., S.R. and T.B. All authors have read and agreed to the published version of the manuscript.

**Funding:** This article has no external funding.

**Institutional Review Board Statement:** Not applicable.

**Informed Consent Statement:** Not applicable.

**Data Availability Statement:** Not applicable.

**Conflicts of Interest:** The authors declare no conflict of interest.

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
