# Peer review of "Recent Advancement of Data-Driven Models in Wireless Sensor Networks: A Survey"

_technologies, doi:10.3390/technologies9040076_

Round 1

Reviewer 1 Report

The authors have made appropriate changes on the paper. I think they should still include a comparison about the data involved in the wireless networks studied in Section 3.

Author Response

Response to Respected Reviewer #1 Comments

Authors are grateful to the reviewer # 1 for his/her positive and encouraging comments

Comment: The authors have made appropriate changes on the paper. I think they should still include a comparison about the data involved in the wireless networks studied in Section 3.

Response: In section 4, “Types of Wireless Sensor Network”, specific application is used for precise data models are described in detail for each WSNs type for the comparison of the different nature of the data.

Reviewer 2 Report

This paper presents a good survey on data-driven models for WSNs. The text is well written and the methodology is correct. In general, it is good work and I have no major concerns in recommending its acceptance, just some minor improvements before the final acceptance, as follows:

  • I suggest the authors to consider avoiding the use of sentences in first person (we, our, us), please;
  • Maybe the authors can enlarge the literature review, particularly including reference to papers published in MDPI journals, including the Sensors Journal. One paper that can be included in the literature review is:

“Cooperation among Wirelessly Connected Static and Mobile Sensor Nodes for Surveillance Applications”, Sensors 2013, 13(10), 12903-12928; https://doi.org/10.3390/s131012903

A quick search in the Sensors Journal can find many more papers that can be added to the literature review and enrich the survey.

  • I suggest the authors slightly change the paper structure of the paper. Eliminate Section 1.1, extract the paragraph above Figure 1 along with this figure and create a new Section 2 with the title “WSN Architecture Overview”, and elaborate more on this subject. For instance, you can include the distinction on hierarchical and non-hierarchical WSN, Clustered WSN, hybrid WSN with static and mobile nodes, etc… The rest of the current text, bellow Figure 1, can be integrated to Section 1.   

Author Response

Response to Respected Reviewer #2 Comments

Authors are grateful to the reviewer # 2 for his/her positive and encouraging comments

Comment: I suggest the authors to consider avoiding the use of sentences in first person (we, our, us), please; Maybe the authors can enlarge the literature review, particularly including reference to papers published in MDPI journals, including the Sensors Journal. One paper that can be included in the literature review is:

“Cooperation among Wirelessly Connected Static and Mobile Sensor Nodes for Surveillance Applications”, Sensors 2013, 13(10), 12903-12928; https://doi.org/10.3390/s131012903
A quick search in the Sensors Journal can find many more papers that can be added to the literature review and enrich the survey.

Response: We have corrected and removed the first person (we, our, us) on whole paper. Also, we have added literature review in more detail in Section 3, mostly considered the MDPI database for quick search from the References no [26] to [31]. Table 1 is also enhanced according to the suggestion. Also added the mention paper as a [12].

Comment: I suggest the authors slightly change the paper structure of the paper. Eliminate Section 1.1, extract the paragraph above Figure 1 along with this figure and create a new Section 2 with the title “WSN Architecture Overview”, and elaborate more on this subject. For instance, you can include the distinction on hierarchical and non-hierarchical WSN, Clustered WSN, hybrid WSN with static and mobile nodes, etc… The rest of the current text, bellow Figure 1, can be integrated to Section 1. 

Response: A new Section 2 with the title “WSN Architecture Overview” has been added and described according to the suggestion by including the hierarchical and non-hierarchical WSN, Clustered WSN, hybrid WSN with static and mobile nodes.

Reviewer 3 Report

The paper presents a survey of data-driven models in wireless sensor networks. In general, the paper is well written and organized. However, there is no contribution to the field. All the information discussed in the survey is well known by the researchers, Section 3 presents different types of WSNs already explored in the literature, including some surveys. The most interesting part is section 4, but again most of this section deals with well-known subjects. Table 2 is interesting, but it is not clear how the authors selected these papers. There are many other works in the area that could be discussed. 

Section 5 (limitation and challenges) is a brief discussion with no novel insights or future research directions. There are many other surveys on WSNs with much more information and useful insights. For example, the authors could also explore more recent aspects of WSNS, such as those related to IoT and LPWANs. I think the paper is not adequate for a journal publication.

Author Response

 Authors are grateful to the reviewer # 3 for his/her positive and encouraging comments

Comment: Table 2 is interesting, but it is not clear how the authors selected these papers. There are many other works in the area that could be discussed.

Response: We have discussed the Table 2 in more detail based on all four data driven model.

Comment: Section 5 (limitation and challenges) is a brief discussion with no novel insights or future research directions. There are many other surveys on WSNs with much more information and useful insights.

Response: We have discussed the future directions more detail based on all four data driven model in the end of section 6 (Limitation and challenges).

Round 2

Reviewer 1 Report

The authors have addressed all my previous comments so I think that the paper is acceptable for publication.

Reviewer 3 Report

The authors included more information in the paper. However, the main issue of the paper (lack of contribution) was not answered by the authors. I repeat the same comments below:

The paper presents a survey of data-driven models in wireless sensor networks. In general, the paper is well written and organized. However, there is no contribution to the field. All the information discussed in the survey is well known by the researchers. Section 3 presents different types of WSNs already explored in the literature, including some surveys. The most interesting part is section 4, but again most of this section deals with well-known subjects. 

This manuscript is a resubmission of an earlier submission. The following is a list of the peer review reports and author responses from that submission.

Round 1

Reviewer 1 Report

  • English needs to be revised by a native speaker. The text contains multiple typos, which makes the reading difficult.
  • The authors should explain some techniques to extend the WSN lifetime, like clustering or scheduling algorithms.
  • Figure 2 is very interesting but it is too small.
  • Classification in Section 3, is it made by the authors or proposed in other works? Please, specify.
  • Why is it Section 3 needed? Is the type of network related to the data-driven method? The authors should discuss about this.
  • Section 4 and Section 5 are the main contributions of this paper. To be a review paper, more works should be analysed and the paper should start focusing on the main properties of each work, and then extracting more features of the implementation analysed. A comparative study concerning their performance could be executed.

Reviewer 2 Report

The work presents a survey on the most recent data-driven approaches to wireless sensor networks.

An initial problem that the work presents is that it does not make clear its research gap. Despite the work having a list of its contributions highlighted in section 1, and being compared to some (few) surveys available on the topic, it is not clear exactly what the article tries to contribute, leaving the idea of dealing, merely, of an organized (but not systematic!) review of articles.

This feeling is reinforced by the reading of the final session of the work, session 5, which is extremely short, leaving the reader with the impression of something unfinished or even absent.

It is possible to see in the text the points that the authors, in the introduction, called "contributions", however, it is difficult, throughout the text, to understand whether these highlights are, in fact, contributions to the area.

In addition, the article presents formatting variations throughout the text, and a need for better quality in the figures provided. In addition, it is necessary to highlight which figures are of the author's own authorship, which is important when it comes to a survey.

I emphasize here that the article has the potential to be a reference work, but it has several structural problems.

Thus, my suggestion is that the text undergo a thorough revision, mainly in terms of form and method, highlighting the importance of the contribution that this survey would have, and making it very clear that this contribution was not just a list of works.